# Mental health and addiction health service use by physicians compared to non-physicians before and during the COVID-19 pandemic: A population-based cohort study in Ontario, Canada

Daniel T. Myran[1,2,3]*, Rhiannon Roberts[1], Eric McArthur[4], Nivethika Jeyakumar[4], Jennifer M. Hensel[5], Claire Kendall[1,2,3,6], Caroline Gerin-Lajoie[7,8], Taylor McFadden[7], Christopher Simon[7], Amit X. Garg[4,9,10,11], Manish M. Sood[1,12,13]‡, Peter Tanuseputro[1,2,3,6]‡

1 Clinical Epidemiology Program, Ottawa Hospital Research Institute, Ottawa, Ontario, Canada, 2 Department of Family Medicine, University of Ottawa, Ottawa, Ontario, Canada, 3 ICES uOttawa, Ottawa Hospital Research Institute, Ottawa, Ontario, Canada, 4 ICES Western, London Health Sciences Centre, London, Ontario, Canada, 5 Department of Psychiatry, University of Manitoba, Winnipeg, Manitoba, Canada, 6 Bruyère Research Institute, Ottawa, Ontario, Canada, 7 Canadian Medical Association, Ottawa, Ontario, Canada, 8 Department of Psychiatry, University of Ottawa, Ottawa, Ontario, Canada, 9 Departments of Medicine, Epidemiology & Biostatistics, Western University, London, Ontario, Canada, 10 London Health Sciences Centre, London, Ontario, Canada, 11 Lawson Health Research Institute, London, Ontario, Canada, 12 Ottawa Hospital Research Institute, Ottawa, Ontario, Canada, 13 Department of Medicine, University of Ottawa, Ottawa, Ontario, Canada

‡ These authors are joint senior authors on this work.
* dmyra088@uottawa.ca

**Academic Editor:** Toshiaki A. Furukawa, Kyoto University Graduate School of Medicine / School of Public Health (current) and Nagoya City University Graduate School of Medical Sciences (at the time of the study), JAPAN

## Abstract

### Background

The Coronavirus Disease 2019 (COVID–19) pandemic has exacerbated mental health challenges among physicians and non–physicians. However, it is unclear if the worsening mental health among physicians is due to specific occupational stressors, reflective of general societal stressors during the pandemic, or a combination. We evaluated the difference in mental health and addictions health service use between physicians and non–physicians, before and during the COVID–19 pandemic.

### Methods and findings

We conducted a population–based cohort study in Ontario, Canada between March 11, 2017 and August 11, 2021 using data collected from Ontario's universal health system. Physicians were identified using registrations with the College of Physicians and Surgeons of Ontario between 1990 and 2020. Participants included 41,814 physicians and 12,054,070 non–physicians. We compared the first 18 months of the COVID–19 pandemic (March 11, 2020 to August 11, 2021) to the period before COVID–19 pandemic (March 11, 2017 to February 11, 2020). The primary outcome was mental health and addiction outpatient visits

**Data Availability Statement:** The data set from this study is held securely in coded form at ICES. While data sharing agreements prohibit ICES from making the data set publicly available, access may be granted by ICES to those who meet pre-specified criteria for confidential access, available at https://www.ices.on.ca/DAS. The full analyses plan (data set creation plan) and underlying analytic code are available through ICES on request, understanding that the computer programs may rely on coding templates or macros that are unique to ICES and are therefore either inaccessible or may require modification.

**Funding:** This study received funding from: The Canadian Institutes for Health Research Operating Grant "Protecting and improving the mental health of physicians during and after the COVID-19 pandemic" (## MS3 – 173107 to DTM, PT, MMS, CK); and grant support from the Academic Medical Organization of Southwestern Ontario (#INN21-002 to AXG). The analyses, conclusions, opinions, and statements expressed herein are solely those of the authors and do not reflect those of the funding or data sources; no endorsement is intended or should be inferred. The study sponsors had no role in the design and conduct of the study; collection, management, analysis, and interpretation of the data; preparation, review, or approval of the manuscript; and decision to submit the manuscript for publication.

**Competing interests:** I have read the journal's policy and the authors of this manuscript have the following competing interests: MS received speaker fees from AstraZeneca and PT is supported by a Physicians' Services Incorporated Graham Farquharson Knowledge Translation fellowship. CGL is a CO-PI on a physician wellness grant from the Mach-Gaennslen Foundation matched by the Ottawa Hospital. TM, CGL, CS – The opinions and conclusions expressed are the writers' own and are not those of the Canadian Medical Association. DTM, RLR, EM, NJ, JH, AXG – These authors have declared that no competing interests exist.

**Abbreviations:** COVID-19, Coronavirus Disease 2019; CPSO, College of Physicians and Surgeons of Ontario; DOLC, date of last contact; MHA, mental health and addiction; OHIP, Ontario Health Insurance Plan; SARS-CoV-2, Severe Acute Respiratory Syndrome Coronavirus 2.

overall and subdivided into virtual versus in–person, psychiatrists versus family medicine and general practice clinicians. We used generalized estimating equations for the analyses.

Pre–pandemic, after adjustment for age and sex, physicians had higher rates of psychiatry visits (aIRR 3.91 95% CI 3.55 to 4.30) and lower rates of family medicine visits (aIRR 0.62 95% CI 0.58 to 0.66) compared to non–physicians. During the first 18 months of the COVID–19 pandemic, the rate of outpatient mental health and addiction (MHA) visits increased by 23.2% in physicians (888.4 pre versus 1,094.7 during per 1,000 person–years, aIRR 1.39 95% CI 1.28 to 1.51) and 9.8% in non–physicians (615.5 pre versus 675.9 during per 1,000 person–years, aIRR 1.12 95% CI 1.09 to 1.14). Outpatient MHA and virtual care visits increased more among physicians than non–physicians during the first 18 months of the pandemic. Limitations include residual confounding between physician and non–physicians and challenges differentiating whether observed increases in MHA visits during the pandemic are due to stressors or changes in health care access.

## Conclusions

The first 18 months of the COVID–19 pandemic was associated with a larger increase in outpatient MHA visits in physicians than non–physicians. These findings suggest physicians may have had larger negative mental health during COVID–19 than the general population and highlight the need for increased access to mental health services and system level changes to promote physician wellness.

## Author summary

### Why was this study done?

- The Coronavirus Disease 2019 (COVID–19) pandemic has had adverse impacts on the mental health of health care workers and the general population.

- There is limited evidence of how changes in health care visits for mental health by physicians during COVID–19 compared to changes in non–physicians.

- Additionally, there is limited evidence on whether there were differences in mental health visits between physicians and non–physicians pre–pandemic.

### What did the researchers do and find?

- We linked registration data for all practicing physicians in Ontario, Canada to health administrative data capturing all outpatient and hospital–based mental health visits between March 2017 and August 2021 and compared changes in visits for physicians and non–physicians pre–and post–pandemic.

- Our study included 41,814 physicians and 12,054,070 non–physicians. Pre–pandemic, after adjusting for age, sex, and place of residence, physicians had comparable overall rates of outpatient mental health and addiction (MHA) to non–physicians but had

higher rates of outpatient MHA visits to psychiatrists and lower rates of outpatient MHA visits to family physicians.

- During the first 18 months of the COVID–19 pandemic, the rate of outpatient MHA visits increased by 23.2% in physicians (888.4 pre versus 1,094.7 during per 1,000 person–years) compared to 9.8% in non–physicians (615.5 pre versus 675.9 during per 1,000 person–years) with much larger increases in virtual care visits in physicians than non–physicians.

### What do these findings mean?

- The findings raise concern that the mental health of physicians may have been more negatively impacted during the COVID–19 pandemic than the general population.

- Alternatively, changes in perceptions of stigma and improved access to health care may have increased use of mental health services for physicians.

- Overall, greater interventions aimed at reducing occupation stressors and workplace mental health promotion are indicated.

## Introduction

The Coronavirus Disease 2019 (COVID-19) pandemic has exacerbated mental health challenges among physicians. Physicians have self-reported high levels of anxiety, depression, burnout, and stress during the COVID-19 pandemic [1–5]. A US multicenter survey of physicians from May to October 2020 ($n$ = 3,128) found that a quarter (25%) had anxiety, a third (30%) were stressed, and almost half reported burnout [4]. In Canada, the Canadian Medical Association's National Physician Health Survey in November 2021 ($n$ = 4,121) found that 59% of survey respondents reported that their mental health had worsened since the onset of the pandemic [5]. During COVID-19, physicians have faced specific stressors, including increased risk of Severe Acute Respiratory Syndrome Coronavirus 2 (SARS-CoV-2) infection [6], shortages in protective equipment [7], changes to workloads, presenteeism (working in poor mental or physical health) [8], and being in situations that invoke moral distress (i.e., when resource shortages prevent one from providing the usual standard of care) [9]. We previously reported a 27% increase in outpatient visits for mental health and addiction (MHA) for Ontario physicians during the first year of the COVID-19 pandemic compared to the year pre-pandemic [10].

While evidence suggests that physicians' mental health has worsened during the COVID-19 pandemic, data suggests that the pandemic has impacted the general population's mental health [11–17]. A study of the US general population found that self-reported mental health worsened as the pandemic continued: with 33.0% of respondents in September 2020 reporting symptoms of anxiety and depression compared to 30.9% in June 2020 [15]. In Canada, a repeated cross-sectional survey found that the percent of individuals aged 18 years and older who screened positive for symptoms of depression, anxiety, or posttraumatic stress disorder increased from 21% in the fall of 2020 to 25% in the spring of 2021 [11]. Importantly, comparing pre-pandemic differences and pandemic changes in physician and non-physician mental

health has 2 primary challenges. First, most evidence is from cross-sectional surveys, which cannot directly compare mental health outcomes before the pandemic to during the pandemic. Consequently, apparent changes in rates of mental health distress may reflect a methodological change (e.g., a different study population or measure) or a COVID-19 impact. Second, few survey studies pre-pandemic [18,19] and none during the pandemic have directly compared MHA of physicians and non-physicians, which limits comparability due to potential differences in sampling, questions posed, and outcomes (e.g., physician surveys measuring burnout versus public surveys reporting depression). Consequently, it is unclear if the worsening mental health among physicians is due to specific occupational stressors, reflective of general societal stressors during the pandemic, or a combination. In addition, it is unclear how physician mental health and use of mental health services compares to the general public, both before and during COVID [18,19].

To address these gaps, we used population-level health administrative data to compare physicians' mental health service use to the general public in Ontario, Canada. These databases capture essentially all outpatient visits to a physician and emergency department visits and hospitalizations through the provinces single-payer universal health system. This approach allowed us to longitudinally compare mental health service use changes in physicians and non-physicians and adjust for differences in age and sex. We specifically focused on differences in rates of MHA visits between physicians and non-physicians pre-pandemic and their changes during the pandemic. In addition, we examined differences in visit type (psychiatry, family medicine), in-person versus virtual visits, and acute care visits, as the COVID-19 pandemic led to policy changes that increased access to outpatient virtual care services by renumerating much broader use of telephone and virtual-based care [20]. Given physician-specific occupational stressors, we hypothesize that increases in MHA visits during the COVID-19 pandemic compared to the pre-pandemic period would be greater for physicians relative to non-physicians.

## Methods

### Study design and setting

We conducted a population-based open cohort study in the province of Ontario, Canada's most populous province ($n$ = 14.7 million in 2020). We included all Ontario residents aged 18 to 105 years eligible for the province's single-payer universal health insurance—Ontario Health Insurance Plan (OHIP) between March 11, 2017 and August 11, 2021. Our study time frame covers the 3 years before the COVID-19 pandemic (March 11, 2017 to March 10, 2020) and the first 18 months of the pandemic (March 11, 2020 to August 11, 2021). We followed the Strengthening the Reporting of Observational Studies in Epidemiology (STROBE) guidelines (see S1 Checklist).

### Data sources

We obtained a list of all physicians registered with the College of Physicians and Surgeons of Ontario (CPSO). Physicians (including residents and fellows) must register with the CPSO to practice medicine in Ontario. CPSO physician registry data were de-identified and linked to health administrative data held at ICES (S1 Text). ICES is an independent, non-profit research institute whose legal status under Ontario's health information privacy law allows it to collect and analyze health care and demographic data, without consent, for health system evaluation and improvement. We obtained data on health outcomes and demographic characteristics from 7 individual-level databases linked using unique encoded identifiers and analyzed at

ICES (S1 Text). Our prospective dataset creation plan is available online (S1 Dataset Creation Plan).

## Population

We assessed cohort eligibility between March 11, 2017 and August 11, 2021. The cohort was derived using monthly intervals; individuals contributed to the cohort for only their eligible months. We identified 43,058 unique physicians alive and living in Ontario during our study. Our final open cohort included 41,814 unique physicians, residents, and fellows (1,244 physicians excluded for being OHIP ineligible, missing the date of last contact (DOLC), or did not access OHIP insured services in the past 5 years). We identified 14,622,440 non-physician Ontario residents aged 18 years and older, alive and living in Ontario during our study. Our final open cohort included 12,054,070 unique non-physicians (1,914,518 excluded for being OHIP ineligible, missing DOLC, or not accessing OHIP insured services in the past 5 years). See S1 Fig for the cohort build.

## Exposures

**Physician status.**   All individuals registered with the CPSO were considered physicians, and all other individuals were considered non-physicians.

**COVID-19 pandemic.**   We considered the COVID-19 pandemic to begin on March 11, 2020 (World Health Organization declaration of the pandemic). For statistical analysis, we divided our study into 36 pre-COVID-19 pandemic monthly intervals (March 11, 2017 to February 11, 2020) and 18 COVID-19 pandemic monthly intervals (March 11, 2020 to August 11, 2021). The first 18 months of the COVID-19 pandemic in Ontario were broadly characterized by 3 "waves" of infections and consequent burden on the health system (March 2020 to May 2020, mid-September 2020 to February 2021, and March 2021 to May 2021) with 2 periods of relatively lower levels of infections (June 2020 to September 2020 and June 2021 to August 2021) [21]. Vaccines in Ontario became available to health care workers in the highest risk settings and residents of long-term care facilities starting in mid-December 2020 with a broader roll out of vaccines to health care workers and the general population starting in March/April 2020 [21].

## Outcomes

Our primary outcome was a composite of outpatient visits (including in-person, virtual care, and telemedicine) with a family physician or psychiatrist related to MHA diagnosis (S2 Text and S1 Table). We defined MHA outpatient visits using previously validated definitions [22]. All outpatient visits to psychiatrists were considered MHA-related visits. We considered visits to a family physician an MHA visit if the visit included suitable MHA diagnostic or fee code. As secondary outcomes, we compared outpatient visits by physician specialty (family medicine versus psychiatry) and visit type (in-person versus virtual care and telemedicine) along with acute care MHA visits. Acute care visits (e.g., ED visits or hospitalizations) for MHA were identified when an ICD-10 diagnostic code from a validated definition was listed as the main or most responsible diagnosis for the visit [22].

## Covariates

We included covariates of individuals' age, sex, place of residence (i.e., living in urban or rural regions), and neighborhood income groups (quintiles using Statistics Canada census data) [23]. Data for the above covariates was complete except for income quintile that was missing for 0.5% of physicians and 0.3% of non-physicians. Physicians were missing data for income

were assigned to the richest income quintile neighborhood and non-physicians with missing data for income were excluded from regression models. We also obtained physician specialty information for all physicians (S2 Table). We included a descriptor of previous mental health history (≥1 MHA outpatient, MHA ED visit, or MHA hospitalizations in the prior 2 years). Data on race and ethnicity was not included as individual-level data are not available at ICES.

## Statistical analysis

We presented the aggregate count of visits and the crude and age-sex standardized rates (standardized using single year groups) per 1,000 person-years for all individuals in the study at monthly intervals. We then analyzed changes in MHA visits by comparing the rates during the first 18 months of the pandemic with the 3 years before the pandemic. Using the individual's monthly interval as the unit of analysis, we conducted Poisson models using generalized estimating equations with an exchangeable covariance structure. The dependent variable was the number of MHA visits (and visit subtypes) for each individual in each period, with an offset for log-follow-up time. Each month we included all eligible physicians and a random 1:1 sample of non-physicians. For all models, we included a binary indicator variable for COVID-19, interpreted as the rate of MHA visits during COVID-19 versus before. We included a binary indicator variable for physician status, interpreted as the pre-pandemic difference in the rate of MHA visits compared to non-physicians. We also included a physician status and COVID-19 interaction, interpreted as the difference in changes between the 2 groups during COVID-19. We adjusted for age (specified as continuous in years), sex, neighborhood income quintile, rural place of residence, and quarters as a proxy for seasonality. As a sensitivity analysis in response to peer-review comments, we repeated our models with age included as cubic spline. These models produced incident rate ratios with 95% confidence intervals. We interpreted confidence intervals that did not cross 1 as a difference between time periods or exposure groups. Previously, we observed a much higher rates of MHA visits in psychiatrists relative to non-psychiatrist physicians [10]. Therefore, we conducted a sensitivity test where we repeated all models excluding psychiatrist physicians. Data was analyzed in SAS version 9.4 (SAS Institute, Cary, North Carolina, United States of America).

## Research ethics

This study was approved by the Health Sciences Research Board at Western University in London, Ontario, Canada. This project was conducted under section 45 of Ontario's Personal Health Information Protection Act, which allows ICES to collect personal health information without consent for the purpose of health system evaluation and improvement, and approved by ICES's Privacy and Legal Office.

## Results

Overall, our study population included 41,814 physicians practicing in Ontario between 2017 and 2021 and 12,054,070 members of the non-physician population of Ontario aged 18 or older (**S1 Fig**). The characteristics of physicians and non-physicians on March 11, 2020 are presented in **Table 1**. For physicians, the mean (SD) age was 47.4 (14.0) years, 22,347 (56.5%) were men, 19,677 (49.7%) lived in the wealthiest quintile neighborhoods in Ontario, and 37,659 (95.1%) lived in an urban region. In contrast, non-physicians were older (mean (SD) age of 50.5 (17.6) years), more likely to be female (51.5%), poorer (20.0% living in the richest quintile neighborhoods in Ontario), and less likely to live in an urban residence (89.4%). On March 11, 2020, a great proportion of physicians, 2,362 (6.0%), than non-physicians, 518,034 (4.7%), had 1 or more psychiatry visits in the 2 years before the start of the pandemic and

**Table 1. Characteristics of physician and non–physicians on March 11, 2020.**

| Variable | Non-physicians $n = 10,972,726^A$ (%/SD) | Physicians $n = 39,583^A$ (%/SD) |
|---|---|---|
| Sex, no. (%) | | |
| Female | 5,650,764 (51.5%) | 17,236 (43.5%) |
| Male | 5,321,962 (48.5%) | 22,347 (56.5%) |
| Age in years, mean (SD) | 50.5 (17.6) | 47.4 (14.0) |
| Age in years, no. (%) | | |
| 18–34 years | 2,558,336 (23.3%) | 8,951 (22.6%) |
| 35–49 years | 2,829,019 (25.8%) | 14,124 (35.7%) |
| 50–64 years | 3,012,559 (27.5%) | 10,950 (27.7%) |
| 65+ years | 2,572,812 (23.5%) | 5,558 (14.0%) |
| Location or home address, no. (%) | | |
| Urban | 9,805,300 (89.4%) | 37,659 (95.1%) |
| Rural | 1,137,900 (10.4%) | 1,721 (4.4.%) |
| Neighborhood income quintile, no. (%) | | |
| 1 (Poorest) | 2,137,349 (19.5%) | 3,169 (8.0%) |
| 2 | 2,185,049 (19.9%) | 4,400 (11.1%) |
| 3 | 2,215,218 (20.2%) | 5,164 (13.1%) |
| 4 | 2,203,732 (20.1%) | 6,967 (17.6%) |
| 5 (Richest) | 2,197,952 (20.0%) | 19,677 (49.7%) |
| One or more psychiatrist visits in past 2 years | 518,034 (4.7%) | 2,362 (6.0%) |
| Psychiatry visits per year, mean (SD)$^B$ | 7.0 (13.2) | 19.2 (30.8) |
| One or more family medicine MHA in past 2 years | 2,377,604 (21.7%) | 6,021 (15.2%) |
| Family medicine MHA visits per year, mean (SD)$^B$ | 4.0 (8.8) | 3.6 (9.8) |
| One or more acute care MHA visit in past 2 years, no (%) | 64,811 (0.6%) | 67 (0.2%) |
| Specialty, no. (%) | | |
| Family medicine | | 18,935 (47.8%) |
| Internal medicine | | 3,611 (9.1%) |
| Surgery | | 4,498 (11.4%) |
| Psychiatry | | 2,120 (5.4%) |
| Anesthesia | | 1,333 (3.4%) |
| Training | | 2,315 (5.9%) |
| Other | | 5,292 (13.4%) |
| Missing | | 1,479 (3.7%) |

$^A$Population not equal to whole study population as individuals could enter and exit the study population.

$^B$Among individuals with 1 or more visits in past year. MHA, mental health and addiction.

physicians who saw a psychiatrist had over twice the mean annual number of psychiatrist visits compared to non-physicians (19.2 versus 7.0) **Table 1.** In contrast, fewer physicians, 6,021, (15.2%) than non-physicians, 2,377,604 (21.7%), had 1 or more family medicine visits in the 2 years before the pandemic with a comparable mean number of annual visits (3.6 in physicians versus 4.0 in non-physicians) **Table 1.** The random sample of the non-physician population taken on March 10, 2020 did not differ by sociodemographic characteristics (age, sex, income, rurality) or co-morbid mental health service use in the past 2 years than the entire non-physician population of Ontario, **S3 Table**.

Rates of overall outpatient MHA visits differed between physicians and non-physicians before and during the COVID-19 pandemic (**Fig 1**). Before the pandemic, the crude rates of

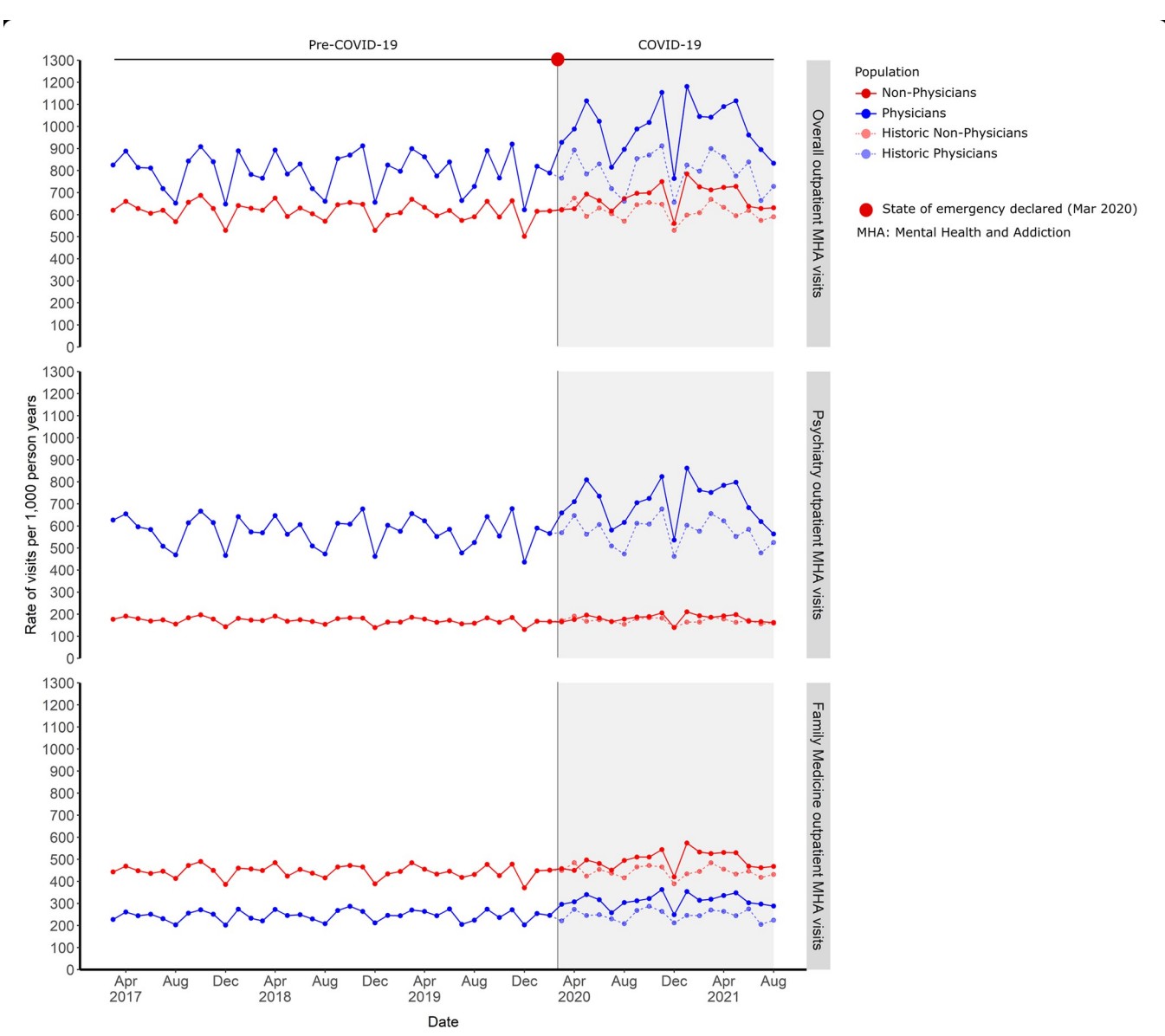

**Fig 1. Age and sex standardized monthly rates of outpatient MHA visits overall, to psychiatrists, and to family physicians among physicians and non–physicians.** Historic rates depict rates from March 2018 to August 2019. COVID–19, Coronavirus Disease 2019; MHA, mental health and addiction.

MHA visits were higher in physicians than non-physicians (888.4 visits per 1,000 person-years in physicians versus 615.5 in non-physicians) (**Tables 2 and S4 for crude IRR**). However, after adjusting for sociodemographic characteristics (age, sex, income, and rurality), this difference was not significant (aIRR 1.00; 95% CI 0.91 to 1.10) (**Table 3**). During the COVID-19 pandemic, there was an increase in the rate of outpatient MHA visits for both physicians and non-physicians compared to pre-pandemic rates. The crude rate of outpatient MHA visits per 1,000 person-years increased by 23.2% in physicians (888.4 pre-pandemic to 1,094.7 during the pandemic) and 9.8% in the non-physician population (615.5 pre-pandemic to 675.9 during the pandemic) (**Table 2**). After adjustment, outpatient MHA visit rates increased for both physicians (aIRR 1.39, 95% CI 1.28, 1.51) and the non-physician population (aIRR 1.12, 95% CI

**Table 2. Rates of MHA visits in physician and non–physicians in the 36 months pre–pandemic (March 11, 2017–March 10, 2020) and during the first 18 months of the pandemic (March 11, 2020–August 11, 2021).**

| | | Pre-COVID-19 pandemic | | During COVID-19 pandemic | | Crude relative change in rate of visits (%) |
|---|---|---|---|---|---|---|
| | | No. visits | Visits per 1,000 person years | No. visits | Visits per 1,000 person years | |
| Overall | Physician | 95,790 | 888.4 | 67,089 | 1,094.7 | 23.2 |
| | Non-physician | 19,816,019 | 615.5 | 11,273,960 | 675.9 | 9.8 |
| Psychiatry | Physician | 65,567 | 608.1 | 45,353 | 740.0 | 21.7 |
| | Non-physician | 5,494,508 | 170.7 | 3,020,394 | 181.1 | 6.1 |
| Family medicine | Physician | 30,223 | 280.3 | 21,736 | 354.7 | 26.5 |
| | Non-physician | 14,321,511 | 444.8 | 8,253,566 | 494.9 | 11.3 |
| In person | Physician | 93,573 | 867.9 | 7,598 | 124.0 | −85.7 |
| | Non-physician | 18,388,655 | 571.1 | 2,737,399 | 164.1 | −71.3 |
| Virtual | Physician | 2,217 | 20.6 | 59,491 | 970.7 | 4,612.0 |
| | Non-physician | 1,427,364 | 44.3 | 8,536,561 | 511.8 | 1,054.5 |
| Acute care | Physician | 329 | 3.1 | 198 | 3.2 | 5.6 |
| | Non-physician | 674,161 | 21.0 | 319,703 | 19.2 | −8.4 |

COVID–19, Coronavirus Disease 2019; MHA, mental health and addiction.

**Table 3. Poisson regression models comparing differences pre–COVID–19 and changes during COVID–19 pandemic in MHA visits between physicians and non–physicians.**

| MHA visit type | Population | Pre-COVID-19 difference | | COVID-19 change (Reference = pre-COVID-19) | |
|---|---|---|---|---|---|
| | | Adjusted incidence rate ratio[A] | 95% CI | Adjusted incidence rate ratio[A] | 95% CI |
| Overall | Physician | 1.00 | (0.91, 1.10) | 1.39 | (1.28, 1.51) |
| | Physician excluding psychiatry | 0.94 | (0.87, 1.00) | 1.44 | (1.37, 1.52) |
| | Non-physician | Reference | [1] | 1.12 | (1.09, 1.14) |
| Psychiatry | Physician | 3.91 | (3.55, 4.30) | 1.22 | (1.16, 1.28) |
| | Physician excluding psychiatry | 2.77 | (2.51, 3.06) | 1.32 | (1.25, 1.40) |
| | Non-physician | Reference | [1] | 1.08 | (1.04, 1.12) |
| Family medicine | Physician | 0.62 | (0.58, 0.66) | 1.33 | (1.25, 1.42) |
| | Physician excluding psychiatry | 0.58 | (0.54, 0.62) | 1.38 | (1.30, 1.46) |
| | Non-physician | Reference | [1] | 1.11 | (1.09, 1.14) |
| Virtual care | Physician | 0.29 | (0.19, 0.44) | 77.03 | (50.64, 117.16) |
| | Physician excluding psychiatry | - | - | - | - |
| | Non-physician | Reference | [1] | 11.69 | (11.14, 12.26) |
| Acute care | Physician | 0.19 | (0.16, 0.23) | 1.03 | (0.81, 1.30) |
| | Physician excluding psychiatry | - | - | - | - |
| | Non-physician | Reference | [1] | 0.90 | (0.82, 0.99) |

[A]Models adjusted for annual quarter, age (continuous), rurality, income–quintile, and sex (male vs. female). Crude IRRs are presented in S4 Table and beta coefficients are presented in S8 Table.

COVID–19, Coronavirus Disease 2019; MHA, mental health and addiction.

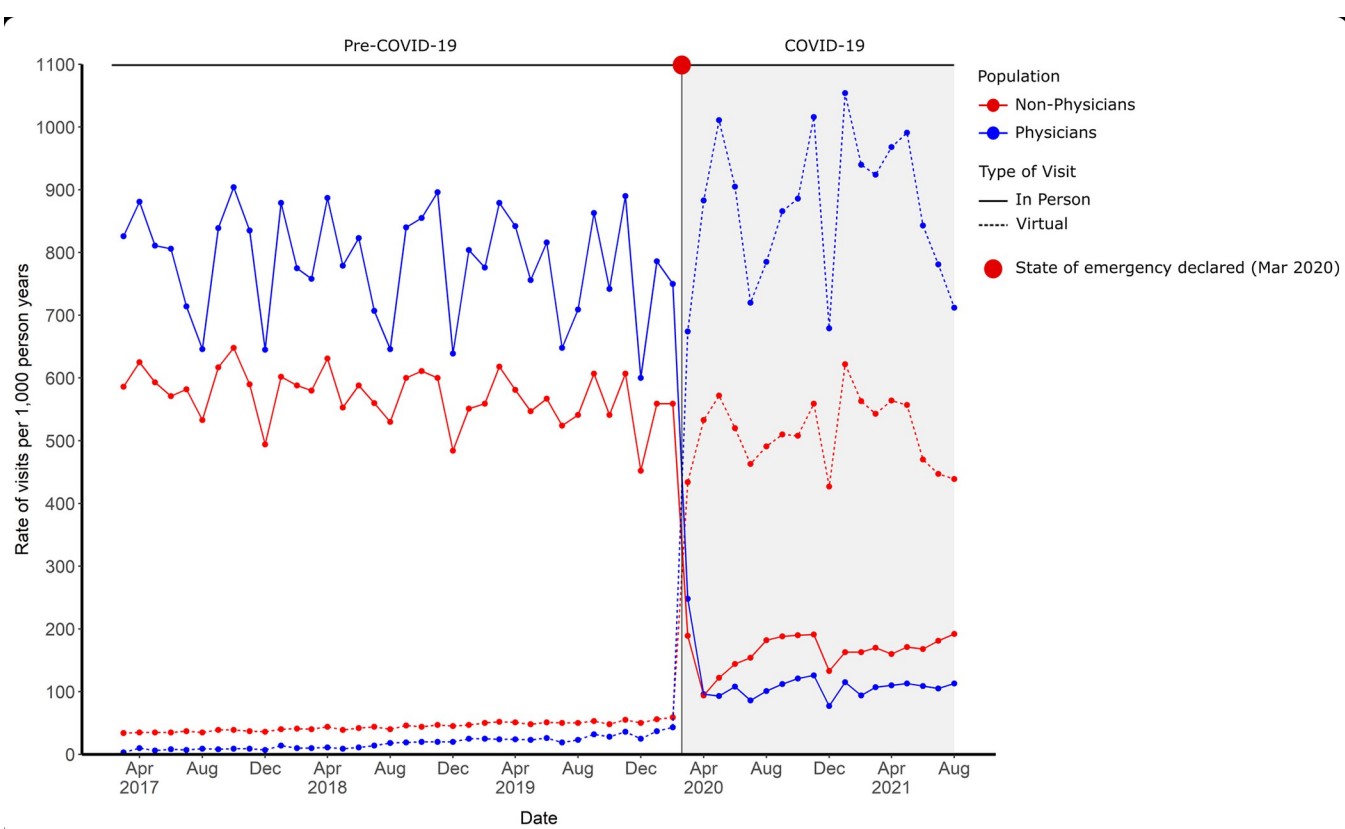

**Fig 2. Age and sex standardized monthly rates of virtual and in–person outpatient MHA visits among physicians and non–physicians.** COVID–19, Coronavirus Disease 2019; MHA, mental health and addiction.

1.09, 1.14) (**Table 3**) and the increase was greater in physicians. Models with age specified as continuous versus cubic spline had comparable results see **S7 Table**.

Physicians and non-physicians had different patterns of MHA visits by provider type. Pre-pandemic, the crude and adjusted rate of outpatient psychiatry visits was over 3 times higher in physicians than non-physicians (aIRR 3.91, 95% CI 3.55, 4.30). Psychiatrists had 8.2 times the rate of outpatient visits to another psychiatrist than the non-psychiatrist physician population (3,584.7 visits per 1,000 person-years versus 437.6) (**S5 Table**). In contrast, pre-pandemic, the crude and adjusted rates of outpatient MHA visits to family physicians were lower in physicians than non-physicians (aIRR 0.62, 95% CI 0.58, 0.66) (**Table 3**). During the pandemic, physicians had a larger increase in MHA visits to both psychiatrists and family physicians than non-physicians.

There was a large increase in outpatient virtual MHA visits for physicians and non-physicians during the COVID-19 pandemic (**Fig 2**). Pre-pandemic rates of virtual outpatient MHA visits in the physician and non-physician population were low. On average pre-pandemic, 2.3% of physician outpatient visits and 7.2% of non-physician outpatient visits were virtual (**Table 2**). In comparison, during the first 18 months of the pandemic, 88.7% of physician outpatient visits and 75.7% of non-physician outpatient visits were virtual (**Table 2**). This increase in virtual care visits was mirrored by large declines of in-person care for both groups (**Fig 2 and Table 2**). During the pandemic, the adjusted increase in rates of virtual MHA visits among physicians (aIRR 77.03, 95% CI 50.64, 117.16) was greater than in non-physicians (aIRR 11.69, 95% CI 11.14, 12.26) (**Table 3**).

Before the COVID-19 pandemic, the crude and adjusted rates of acute care visits were much lower in physicians (3.1 per 1,000 person-years) than non-physicians (21.0 per 1,000 person-years, aIRR 0.19, 95% CI 0.16, 0.23) (**Tables 2 and 3**). During COVID-19, changes in acute care visits did not differ between physicians (aIRR 1.03 95% CI 0.81, 1.30) and non-physicians (aIRR 0.90 95% CI 0.92, 0.99).

The types of outpatient MHA visits by physicians and non-physicians are in **S6 Table**. In both periods, over 65% of MHA visits in physicians were for anxiety, and approximately 15% were for a mood disorder. A much higher proportion of outpatient MHA visits were related to drug and alcohol use in non-physicians (approximately 20% of visits) than physicians (approximately 5% of visits).

## Discussion

Prior research on physician mental health before and during the COVID-19 pandemic has been limited by small study sizes, reliance on self-reported surveys rather than measuring actual health care visits, and lack of comparison to the general population as a control for the society-wide impacts of the pandemic. To our knowledge, this is the largest study to examine the use of mental health services by physicians and non-physicians. Using 5 years of population-level administrative health data for more than 12 million individuals, we found substantial differences in mental health service use between physicians and non-physicians before the COVID-19 pandemic. Further, we found that the COVID-19 pandemic was associated with a much larger increase in physicians' MHA visits than non-physicians.

We found that pre-pandemic physicians had similar rates of MHA outpatient visits relative to non-physicians after accounting for differences in sociodemographic characteristics. Previous studies have found that physicians' self-reported mental health and well-being (e.g., burn-out and work-life dissatisfaction) are worse compared to non-physicians [18,19]. This discrepancy may be explained by physicians' reluctance to seek care due to stigma, confidentiality concerns, fear of licensing implications, and impacts on their career development/progression [24–26]. In addition, a culture of self-treatment [27,28] and a preference to seek help from professionals that are not physicians (e.g., psychologists) [24] may contribute to physicians' underuse of outpatient mental health services. Interestingly, we found physicians had much higher psychiatry visits rates than non-physicians and lower primary care visit rates that persisted after excluding psychiatrists (whom we previously documented access psychiatry services at very high rates) [10]. This finding is likely partially explained by different access to specialized psychiatric services or access through colleagues. Finally, we found a much higher proportion of outpatient MHA visits were related to drug and alcohol use in non-physicians (approximately 20% of visits) than physicians (approximately 5% of visits). This finding may reflect lower levels of substance use among physicians or a reluctance of physicians to seek help for addictions services due to fear of licensing implications and stigma [26].

MHA visits increased in both physicians and non-physicians during the COVID-19 pandemic, consistent with reports of worsening mental health in both groups [1–5,11–16].

However, physicians had a larger increase in MHA visits than non-physicians and in prior work, we showed that increases in MHA visits in physicians during the pandemic were driven both by greater incident mental health visits and greater intensity of mental health service use for those with preexisting mental health conditions [10]. There are a few possible reasons for these findings. First, these findings likely reflect a higher mental health burden on physicians than non-physicians due to occupational stressors. The Canadian Medical Association National Physician Health Survey found the most frequent occupational stressors contributing to worsening mental health among physicians during COVID were increased workload/lack

of work-life integration, longer periods of social isolation, rapidly changing policies/processes, and lack of human resources [29]. Second, these findings may reflect better physician access to mental health care services during the pandemic, particularly outpatient psychiatric services. Third, these findings may be due to greater reductions in barriers to accessing mental health care services for physicians compared to non-physicians. The expanded virtual care options [20] may have allowed physicians with MHA issues predating COVID-19 to access services easily (e.g., more flexible timing and appointment location). Indeed, the uptake of virtual visits was higher among physicians than in the general population. Finally, these findings may also be due to a reduction in mental health stigma among physicians due to the COVID-19 pandemic. During COVID-19, there was frequent messaging by governments and physician organizations concerning the mental health impacts of the pandemic. It is possible that conversations regarding physician mental health during the pandemic reduced stigma and increased treatment access.

While the COVID-19 pandemic has exacerbated physicians' mental health concerns, high levels of burnout predated the pandemic. Our research suggests that an important intervention to help improve physician health and access to mental health services would be to continue virtual mental health services. The pandemic has also placed considerable additional strain on the health system. We suspect the physician mental health burden will continue after the COVID-19 pandemic as physicians work through critical backlogs (e.g., surgical and cancer care) with sicker patients and increase workloads and staffing shortages—as physicians reduce their practice hours or leave the profession [30–32]. Consequently, ongoing surveillance on physician mental health during later stages of the pandemic and recovery is indicated. Importantly, our findings suggest a pressing need for interventions to mitigate physician burnout and mental health concerns. A wide variety of interventions including increasing treatment access and system-level changes aimed at improving work-life balance such as reducing the stress and administrative burden of physician work have been proposed and will require a substantial increase in health system investment [3]. Finally, there should be further research into the patterns of mental health service use by physicians to understand care continuity and quality.

## Strengths and limitations

The strengths of our study include a large sample size (>40,000 physicians), a group for comparison (>12,000,000 general population), a longitudinal design, and an analytic approach to control for pre-pandemic trends. Our study uses data from a universal public health insurance plan that captures almost all outpatient physician visits, ED visits, and hospitalizations. Collectively, these strengths support high generalizability of our findings to other physician populations. However, our study does have some limitations. First, this study cannot disentangle whether the greater increases in MHA services during the COVID-19 pandemic for a physician reflects worsening mental health or better access to mental health services. Second, while we adjusted for age, sex, rurality, and income, we expect large residual differences in socioeconomic status between physicians and non-physicians. Third, our study does not capture mental health services delivered by non-physician professionals, which are not captured in our datasets. Some types of physicians (e.g., younger) may prefer the services of psychologists or other providers to those provided by physicians [24]. Therefore, our study may be underestimating increases in both physician and non-physician use of mental health care services.

## Conclusions

During the first 18 months of the COVID-19 pandemic, physicians had a greater increase in outpatient MHA visits relative to the general population. These findings suggest that the

mental health of physicians may have been more negatively impacted by COVID-19 than the general population. Interventions aimed at reducing occupation stressors and workplace mental health promotion are indicated. Continuation of virtual care options will likely enable better access and utilization of care when needed.

## Supporting information

**S1 Checklist. STROBE Statement—Checklist of items that should be included in reports of cohort studies.**
(DOCX)

**S1 Fig. Study cohort build.**
(DOCX)

**S1 Dataset Creation Plan. Study analysis plan.**
(DOCX)

**S1 Text. Data sources, outcome/covariate/exposure, and cohort descriptions.**
(DOCX)

**S2 Text. Definition of MHSU visits and type of visit.**
(DOCX)

**S1 Table. List of OHIP codes.**
(DOCX)

**S2 Table. List of all physician specialities.**
(DOCX)

**S3 Table. Characteristics of all non–physicians in Ontario on March 10, 2020 and random sample of non–physicians taken on March 10, 2020.**
(DOCX)

**S4 Table. Crude Poisson regression models comparing differences pre–COVID–19 and changes during COVID–19 pandemic in mental health and addiction visits between physicians and non–physicians.**
(DOCX)

**S5 Table. Rates of outpatient health and addiction visits in psychiatrist physicians and non–psychiatrist physicians in the 36 months pre–pandemic (March 2017–February 2020) and during the first 18 months of the pandemic (March 2020–August 2021).**
(DOCX)

**S6 Table. Counts of outpatient mental health and addiction–related codes by physicians during the first 18 months of the COVID–19 pandemic compared 3 years before the pandemic.**
(DOCX)

**S7 Table. Poisson regression models comparing differences pre–COVID–19 and changes during COVID–19 pandemic in mental health and addiction visits between physicians and non–physicians.**
(DOCX)

**S8 Table. Adjusted Poisson regression models comparing differences pre–COVID–19 and changes during COVID–19 pandemic in mental health and addiction visits between**

physicians and non–physicians.
(DOCX)

## Author Contributions

**Conceptualization:** Daniel T. Myran.

**Formal analysis:** Daniel T. Myran, Eric McArthur, Nivethika Jeyakumar.

**Funding acquisition:** Daniel T. Myran, Manish M. Sood, Peter Tanuseputro.

**Investigation:** Nivethika Jeyakumar.

**Methodology:** Daniel T. Myran, Rhiannon Roberts, Eric McArthur.

**Project administration:** Rhiannon Roberts.

**Supervision:** Manish M. Sood, Peter Tanuseputro.

**Writing – original draft:** Daniel T. Myran, Rhiannon Roberts.

**Writing – review & editing:** Daniel T. Myran, Rhiannon Roberts, Eric McArthur, Nivethika Jeyakumar, Jennifer M. Hensel, Claire Kendall, Caroline Gerin-Lajoie, Taylor McFadden, Christopher Simon, Amit X. Garg, Manish M. Sood, Peter Tanuseputro.

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
