## [Editor Report · Decision Letter 0]

14 Jul 2022

Dear Dr Myran, 

Thank you for submitting your manuscript entitled "Outpatient mental health and addiction visits by physicians compared to non-physicians before and during the COVID-19 pandemic." for consideration by PLOS Medicine.

Your manuscript has now been evaluated by the PLOS Medicine editorial staff and I am writing to let you know that we would like to send your submission out for external peer review.

Your manuscript is within the scope of our upcoming Special Issue on the COVID-19 pandemic and global mental health (https://speakingofmedicine.plos.org/2022/04/05/plos-medicine-special-issue-the-covid-19-pandemic-and-global-mental-health/). The deadline for the Special Issue is December 15 2022, with anticipated publication in Q1 2023 (subject to change dependent on submission volume). We intend to publish all papers accepted for the Special Issue simultaneously.

When revising your manuscript for full submission, please indicate in the cover letter whether you would like to have your manuscript considered for the special issue (note that your manuscript will still be sent for review even if you would prefer not to be part of the special issue). If you have any other questions, please feel free to contact me (cdavidson@plos.org).

Before we can send your manuscript to reviewers, we need you to complete your submission by providing the metadata that is required for full assessment. To this end, please login to Editorial Manager where you will find the paper in the 'Submissions Needing Revisions' folder on your homepage. Please click 'Revise Submission' from the Action Links and complete all additional questions in the submission questionnaire.

Please re-submit your manuscript within two working days, i.e. by Jul 18 2022 11:59PM.

Kind regards,

Callam Davidson

Associate Editor

PLOS Medicine

---

## [Decision Letter · Decision Letter 1]

12 Dec 2022

Dear Dr. Myran,

Thank you very much for submitting your manuscript "Outpatient mental health and addiction visits by physicians compared to non-physicians before and during the COVID-19 pandemic." (PMEDICINE-D-22-02286R1) for consideration at PLOS Medicine. 

Your paper was evaluated by an associate editor and discussed among all the editors here. It was also discussed with an academic editor with relevant expertise, and sent to independent reviewers, including a statistical reviewer. The reviews are appended at the bottom of this email and any accompanying reviewer attachments can be seen via the link below:

[LINK]

In light of these reviews, I am afraid that we will not be able to accept the manuscript for publication in the journal in its current form, but we would like to consider a revised version that addresses the reviewers' and editors' comments. Obviously we cannot make any decision about publication until we have seen the revised manuscript and your response, and we plan to seek re-review by one or more of the reviewers. 

We hope to receive your revised manuscript by Jan 09 2023 11:59PM. Please email us (plosmedicine@plos.org) if you have any questions or concerns.

We look forward to receiving your revised manuscript. 

Sincerely,

Callam Davidson, 

PLOS Medicine

plosmedicine.org

Please revise your title according to PLOS Medicine's style: please place the study design ("A population-based cohort study") in the subtitle (ie, after a colon). To ensure clarity, we suggest "Outpatient mental health and addiction service use by physicians compared to non-physicians before and during the COVID-19 pandemic: a population-based cohort study", or similar.

Please include continuous line numbering throughout the manuscript to facilitate further review.

Abstract Methods and Findings: Please update the term ‘COVID-19 pandemic period’ to ‘the first 18 months of the COVID-19 pandemic’ or similar. 

The Data Availability Statement (DAS) requires revision. 

* The URL leads to an error page, please check and update as required. 

* You state that ‘The full data set creation plan and underlying analytic code are available from the authors on request’. Please note that a study author cannot be the contact person for the data. See our website for suggested alternatives: https://journals.plos.org/plosmedicine/s/data-availability

The funding information can be removed from the cover page – this information is captured via the submission form and will be published as metadata in the event of acceptance.

Please place citations within square brackets and preceding punctuation throughout your manuscript.

When citing your Supporting Information (e.g., Methods Supplement), please use the naming conventions outlined here: https://journals.plos.org/plosmedicine/s/supporting-information

Please define the abbreviation MHA in the Figure 1 legend.

Please provide the crude IRRs as well as the adjusted IRRs in Table 3 (or alternatively please report the crude analyses in the Supporting Information and cite these in the Results).

Please ensure that the study is reported according to the STROBE guideline, and include the completed STROBE checklist as Supporting Information. Please add the following statement, or similar, to the Methods: "This study is reported as per the Strengthening the Reporting of Observational Studies in Epidemiology (STROBE) guideline (S1 Checklist)."

Did your study have a prospective protocol or analysis plan? Please state this (either way) early in the Methods section.

Comments from the reviewers:

Reviewer #1: Thanks for the opportunity to review your manuscript. My role is as a statistical reviewer, so my review concentrates on the study design, data, and analysis that are presented. I have put general questions first, followed by queries relevant to a specific section of the manuscript (with a page/paragraph reference).

This study is an observational cohort study that compares changes in utilisation of healthcare between physicians and general population from 2017 to 2021, specifically family medicine and outpatient mental health and addiction visits. The study uses routinely collected health data, linked to physician/surgeon registration data. The study uses a difference-in-difference approach, estimating the difference between physicians and non-physicians in the difference between pre-COVID and COVID time periods (done by including an interaction between a COVID time period indicator and physician indicator variable). 

One of the strengths of the study is the size and coverage - the integrated data available from this province is impressive and I think is a demonstration of the value of building data infrastructure as this study would have been impossible to do prospectively. 

It wasn't directly specified in the methods, but it looks as though age-groups included were relatively large (e.g. 35-49, 50-64). In terms of the effect of reducing confounding, this assumes that a 35 year old has the same levels of healthcare utilisation as a 49 year old. I would like to see a sensitivity analysis that includes age as a continuous variable in spline form to see if this reduces the differences between physicians and non-physicians as I suspect the age structure between the two groups within the 35-49 and the 50-64 age-groups could be different. 

How many (%) of the cohort (non-phys/phys) had missing information on neighbourhood income quintile? Were those missing this information different to those with complete data? 

P8, Paragraph 2. Is the form of the covariates used in the main analyses the same as in table 1? 

P8, Paragraph 3. What age-groups were used for the age standardisation? 

P9, Paragraph 1. What type of covariance structure (e.g. exchangeable, AR(1)) was used in the GEEs? 

Was the suitability of the Poisson distribution assessed? E.g. any evidence of excess 0s, or overdispersion?

Was autocorrelation checked for?

Why was a random sample of non-physicians used instead of just including all available data? 

What criteria were you using to infer that there is a difference between time-periods or exposure groups? 95% CI? 

P11, Table 1. I would add a 'No. (%)' to the age part of the table where this is presented by age-group. 

P12 Paragraph 1. Can the regression results (e.g. beta coefficients) of these regression models used to estimate IRR be included as an appendix? 

P16. Paragraph 1. The change is virtual visits is enormous - were there changes in policy at the time to allow more patients to use these, or to use these instead of in-person visits where possible?

eTable 1. What is the category of the top row of each variable? Is this the missing data for that variable? 

Reviewer #2: Mental health help seeking among physicians, particularly any changes over the pandemic, is an important topic that is often discussed without research to back it and this paper is the first to do so. The use of a population based cohort and confirming physician status by registration, helps to limit any lack of disclosure of job title by physician's themselves for fear of stigma and creates a large cohort.

A few suggestions to help improve upon an already important paper:

- Introduction:

 - The paragraph on p.5 reads like a limitations paragraph instead of introductory paragraph and could be reworded to emphasize what you did, in the context of the limitations of prior work, instead of starting with the limitations. 

 - Is there anything unique about the pandemic in ontario that might have influenced the population as a whole, but physicians specifically, that readers need to know? Anything that might make the population any different from physicians anywhere else? 

 - It might be beneficial to explain to non-canadian readers what access to care for mental health through the health system looks like and if there is any difference for a physician or not. In the U.S, the argument might be there is more access to care for physicians than non-physicians but as that access to care is through their employer, or employee access plans, perhaps many people go outside of the system to not be seen in relation to work and it would be hard to track. This is also especially important as later you say psychiatry might be used more by physicians due to access to specialty care.

Methods: Appreciated the large sample and controlling for specific variables. Would wonder if there was any lag in the increase over the pandemic time period for physicians seeking help and if a lot of it has come in more recent months. Would also wonder if there was any decrease around times of high covid spikes and whether that was even the opposite of the non-physicians (non physicians had more need, got more help, physicians had the need but had to help others). This might be outside the scope of the paper, but many people have argued that there was a delay due to physicians having to focus on taking care of others (even with increasing access).

Results/Discussion:

- Why do the authors think that physicians sought care at higher numbers in comparison to non-physicians even pre-pandemic? Yes, this negligible with adjusted rates, but even being the same as the general population seems like it would be unexpected. 

- Why do the authors think psychiatrists had higher rates of using psychiatrists? Is it simply that they are psychiatrists? Was there any difference pre/post pandemic in this group that would explain carrying the burden of this increase in people needing psychiatry over this time? This seems worth mentioning as to sustain care for the population, psychiatrists need to be mentally well. Even just mentioning that in the discussion seems important

- p.18: Is it possible physicians lie about their substance use so it isn't coded as a problem? Especially because of licensing concerns? Is anxiety more acceptable of a diagnosis?

-p.19: Did Ontario/Canada do anything specific to change access to care for physicians over the pandemic that might make access explain the increase? Yes, telehealth helped, but that should've helped all groups. Some hospital systems increased their programs for mental health in healthcare workers over the pandemic and if this increased use/care it is worth mentioning as then this is worth the investment/sustainability (and something to say in discussion/conclusion). 

- p.19: In the conversation about reduction of barriers, what about actual anti-stigma, normalization campaigns by hospital systems and more awareness/acceptance of need in this group? Care seeking might increase if it is talked about more even through social media/the news and this is worth mentioning as again, then it could be worth investing in and sustaining. Did Ontario/Canada do anything specific in this realm worth mentioning?

- p.19- You discuss a systems investment as part of a solution, but what systems level investment is needed to sustain an increasing need/access to care and continue to decrease barriers to care? Given the focus of the paper, consider things like opt-out screening, COPE columbia program, as a guide. This seems more important to focus on as a conclusion/take home than the systems causes of mental health problems to the focus of your paper.

Thank you for your work!

Reviewer #3: Thank you for allowing me to review this revision. This study looked at psychiatry visits amongst physicians compared to non-physicians during the COVID-19 pandemic. Overall an important topic with significant implications for the healthcare workforce. Comments below:

-unclear if Canadian system all care is accessed and readily available in this database searched. For example in the US many providers may pay cash to avoid the "stigma" of having a mental health visit notched in their record. Could this have been a possibility in this sample? 

-With COVID-19 dates, were volumes similar to other countries during that time? I know 3/11 was the WHO declaration but how was the situation in Ontario? Has implications on the duration of the proposed "exposure" (e.g covid stress) that the providers were or not facing. May have varied geographically within the province as well. 

-why only family medicine and psych as the speciality? What about other specialities with whom covid care burden would have been high during the initial phase or surges of the pandemic (e.g critical care, emergency medicine). 

-can you discuss a bit more about implications for this work and efforts aiming to destigmatize mental health amongst clinicians 

Reviewer #4: Thank you for giving me the opportunity to review this manuscript. Mental health problem is an important problem among physicians during COVID-19 pandemic. The topic of this manuscript is interesting and have clinical implementation. However, there are some concern for this manuscript. 

1. The first concern is that as a population-based cohort study, the author should give the flow chart to describe the maintenance information and lost to follow-up of the cohort etc. 

2. The author did not present the detailed demographic information, did not distinguish the mental health history and new onset of mental health information. The mental health visit information mixed by many other confounders. 

3. The current mental health visit information analysis could be aided by the analysis of prescribing of psychiatric drugs. The authors are recommended to add the information of antidepressant, or antipsychiatry drugs pre pandemic and during COVID-19.

[LINK]

---

## [Decision Letter · Decision Letter 2]

17 Jan 2023

Dear Dr. Myran,

Thank you very much for re-submitting your manuscript "Outpatient mental health and addiction service use by physicians compared to non-physicians before and during the COVID-19 pandemic: a population-based cohort study" (PMEDICINE-D-22-02286R2) for review by PLOS Medicine.

I have discussed the paper with my colleagues and the academic editor and it was also seen again by three reviewers. I am pleased to say that provided the remaining editorial and production issues are dealt with we are planning to accept the paper for publication in the journal.

[LINK]

As your manuscript is under consideration for our Special Issue, by default it will be opted out of the early version process and an uncorrected proof of your manuscript will not be published online ahead of the final version. If you would like to opt in to the early versioning process or discuss this option further, please contact the editor (cdavidson@plos.org). 

We look forward to receiving the revised manuscript by Jan 24 2023 11:59PM.   

Sincerely,

Callam Davidson, 

Associate Editor 

PLOS Medicine

plosmedicine.org

Requests from Editors:

Please include the additional sensitivity analyses conducted in response to Reviewer 1 in the Supporting Information and reference them in the Methods and Results.

Related to the above, changes in the analysis-- including those made in response to peer review comments-- should be identified as such in the Methods section of the paper, with rationale.

Please also include the beta coefficients (Reviewer 1 Comment 11) in the Supporting Information.

Please also include the flow chard (Reviewer 4, Comment 1) in the Supporting Information.

Please include the study setting (Ontario) in the title. 

Please remove the ‘Disclosures’ from the title page and the ‘Funding’ section from the end of the manuscript (these are captured as metadata via the Submission Form so please ensure all relevant information is captured on resubmission).

In the last sentence of the Abstract Methods and Findings section, please describe the main limitation(s) of the study's methodology.

Please ensure all citations precede punctuation (e.g., line 152).

Please cite all items in the Supporting Information according to our guidelines: https://journals.plos.org/plosmedicine/s/supporting-information

Related to the above, please ensure all items in the Supporting Information are numbered correctly (e.g., possible issues at lines 214, 260, 311, and 344).

Please include the numerator and denominator when reporting percentages (e.g., lines 266 and 269). 

Please ensure all abbreviations used in Tables/Figures are defined in the respective legends.

Please enlarge the text used in the key/axis labels in Figures 1 and 2. 

Please provide the date of citation for internet sources (e.g., References 11 and 22).

Comments from Reviewers:

Reviewer #1: Thanks for the revised manuscript and responses to my review. Apologies for not picking up that age was included as a continuous variable. Including the sensitivity analysis as a supplement would be fine with me as the results are very similar to including it in the linear form. I agree that use of the Poisson model to estimate RR here with individual level counts of health service utilisation should be fine and check of residuals is not needed. The revisions and responses to my queries have resolved my initial review. I recommend that the flowchart produced at R4's request be included as an appendix for the study. 

Reviewer #2: Thank you for your work to improve upon your manuscript based on the reviews. Really appreciate the contextualizing of the data in time and location.

My only question is that physicians seeking care at higher numbers pre-pandemic, doesn't align solely with there being greater need. There has always been high need in physicians, higher than other groups, but due to systemic and cultural factors, they don't seek care and studies have shown that they seek care in low numbers. Is there another explanation for this service usage? Or, can you say that it is not exclusively because of the higher need/rates and more studies are indicated to understand more?

Reviewer #3: The authors have addressed my comments and I have no additional comments. Thank you very much to the author team!

[LINK]

---

## [Editor Report · Decision Letter 3]

26 Jan 2023

Dear Dr Myran, 

On behalf of my colleagues and the Academic Editor, Professor Toshiaki Furukawa, I am pleased to inform you that we have agreed to publish your manuscript "Mental health and addiction health service use by physicians compared to non-physicians before and during the COVID-19 pandemic: a population-based cohort study in Ontario, Canada" (PMEDICINE-D-22-02286R3) in PLOS Medicine.

When making the formatting changes, please also make the following editorial changes:

* Please update your title to 'Mental health and addiction health service use by physicians compared to non-physicians before and during the COVID-19 pandemic in Ontario, Canada : a population-based cohort study'

* Line 303: Please correct to 'greater proportion'.

PRESS

We ask that you take this opportunity to read our Embargo Policy regarding the discussion, promotion and media coverage of work that is yet to be published by PLOS. As your manuscript is not yet published, it is bound by the conditions of our Embargo Policy. Please be aware that this policy is in place both to ensure that any press coverage of your article is fully substantiated and to provide a direct link between such coverage and the published work. For full details of our Embargo Policy, please visit http://www.plos.org/about/media-inquiries/embargo-policy/.

SPECIAL ISSUE PUBLICATION

We are intending to publish the Special Issue in April 2023. We appreciate that some authors may wish to begin publicising their work before the Special Issue launches. For this reason, we have decided to offer optional Early Article Posting for authors who feel that this would be beneficial.

Posting an early version of the article online removes the press embargo and allows authors to begin coordinating their own/institutional press activities, though it is important to note that this can result in reduced media coverage overall (we have published a related article on this in the context of preprints: https://theplosblog.plos.org/2020/05/preprints-and-the-media-a-change-to-how-plos-handles-press-for-papers-previously-posted-as-preprints/). 

If you feel that your article may benefit from early versioning, please reach out to me (cdavidson@plos.org) and I will be happy to discuss this further. If we do not hear otherwise, we will assume you are happy to remain opted out of the early versioning process.

REPRODUCIBILITY

Sincerely, 

Callam Davidson 

Associate Editor 

PLOS Medicine